



# A Sudden Stratospheric Warming Compendium

Amy H. Butler[1,2], Jeremiah P. Sjoberg[1,2], Dian J. Seidel[3], and Karen H. Rosenlof[2]

[1]Cooperative Institute for Research in Environmental Sciences, University of Colorado, Boulder, CO, 80309, USA
[2]National Oceanic and Atmospheric Administration, Earth Systems Research Laboratory, Chemical Sciences Division, Boulder, CO, 80305, USA
[3]National Oceanic and Atmospheric Administration, Air Resources Laboratory, College Park, MD, 20740, USA (retired)

*Correspondence to*: Amy H. Butler (amy.butler@noaa.gov)



**Abstract.** Major mid-winter sudden stratospheric warmings (SSWs) are large and rapid temperature increases in the wintertime polar stratosphere associated with a complete reversal of the climatological westerly winds (i.e., the polar vortex). These extreme events can have substantial impacts on wintertime surface climate, such as cold air outbreaks over North

America and Eurasia, or anomalous warming over Greenland. Here we present a SSW Compendium (SSWC), a new database that documents the evolution of the stratosphere, troposphere, and surface conditions 60 days prior to and after SSWs for the period 1958-2014. The SSWC comprises data from six different reanalysis products: MERRA2 (1980-2014), JRA-55 (1958-2014), ERA-interim (1979-2014), ERA-40 (1958-2002), NOAA20CRv2c (1958-2011), and NCEP-NCAR I (1958-2014). Global gridded daily anomaly fields, full fields, and derived products are provided for each SSW event. The

compendium will allow users to examine the structure and evolution of individual SSWs, and the variability among events and among reanalysis products. The SSWC is archived and maintained by NOAA's National Centers for Environmental Information (NCEI, doi: http://dx.doi.org/10.7289/V5NS0RWP).

**Keywords**. Stratosphere, climate variability, ozone, potential vorticity, sudden stratospheric warming, stratosphere-
troposphere coupling, polar vortex

## 1 Introduction

The wintertime polar stratosphere is highly dynamic. In the Northern Hemisphere (NH), breaking planetary-scale waves propagating up from the troposphere or the excitation of resonant modes can lead to the disruption and deceleration of the climatological westerly circulation of the polar vortex (see Schoeberl 1978 for a historical review). Associated with this

wind deceleration is a dramatic warming, sometimes increasing the temperature of the polar stratosphere by as much as 30-40 degrees K in a few days. In the most extreme cases, the stratospheric polar vortex can reverse direction completely, in an event called a major "sudden stratospheric warming" (SSW). SSWs in the NH occur roughly 6 times per decade (Charlton and Polvani, 2007). SSWs can also occur in the Southern Hemisphere (SH), as in a remarkable case in September 2002 (Kruger et al., 2005), but are rare due to smaller planetary wave amplitudes in the SH (van Loon et al., 1973).

Large perturbations in the stratospheric circulation can drive changes in surface climate for days to weeks (Kidston et al., 2015). In particular, SSWs are often followed by an equatorward shift of the North Atlantic tropospheric storm track, projecting onto the spatial pattern of the negative phase of the North Atlantic Oscillation (NAO). On average, this pattern results in warm anomalies over Greenland, eastern Canada, and subtropical Africa and Asia, and cold anomalies over northern Eurasia and the eastern United States. However, the impacts of individual SSWs vary widely, depending on the

evolution of the vortex breakdown, the strength of the stratospheric-tropospheric coupling, and the state of the tropospheric climate.





Because of the impact of SSWs on wintertime surface climate and mid-latitude cold air outbreaks, as well as their potential influence on ozone and chemical transport (e.g., Manney et al., 2009; Schoeberl and Hartmann, 1991), tropical convection and dynamics (e.g., Gómez-Escolar et al., 2014; Kodera, 2006), and mesospheric processes (e.g., Hoffmann et al., 2007), a "research-ready" database of these events would be useful. Daily three-dimensional gridded variables are

needed to examine the full evolution and impacts of SSWs. Therefore reanalysis products, which assimilate observations to constrain a global climate model, are often used. But the calculation of daily anomalies or additional derived products using reanalysis data can be computationally expensive and storage intensive. In addition, different reanalyses also differ in time spans, assimilated observations, assimilation scheme, parameterizations, and model physics, so intercomparison of multiple reanalysis products is useful to assess what features of SSWs and their associated climate variability are robust.

Here we describe a SSW Compendium (SSWC), which provides a detailed historical dataset of major SSWs, allowing users to consider the development, evolution, and impacts of individual SSWs and to provide a basis for model evaluation and improvement. A "compendium" is a concise compilation of comprehensive information on a specific subject, and therefore is an appropriate term to describe this dataset. The SSWC includes data from six established reanalysis products, and includes anomaly fields and additional derived products to highlight the dynamics and effects of SSW events.

We present an overview of the reanalysis source data and the methodology for SSW event selection and data processing in Section 2. Section 3 discusses potential applications of this database, and Section 4 highlights the availability of the database at the National Oceanic and Atmospheric Administration (NOAA) National Centers for Environmental Information (NCEI) archives and at the NOAA Earth Systems Research Laboratory (ESRL).

## 2 Methodology

### 2.1 Reanalysis data

The SSWC comprises data from six different reanalyses (**Table 1**): the National Aeronautics and Space Administration (NASA) Modern-Era Retrospective-analysis for Research and Applications version 2 (MERRA2), Japanese 55-year Reanalysis (JRA-55), European Centre for Medium-Range Weather Forecasts (ECMWF) 40-year Reanalysis (ERA-40), ECMWF Interim Reanalysis (ERA-interim), NOAA 20[th] Century Reanalysis version 2c (NOAA20CRv2c), and

NOAA's National Centers for Environmental Prediction/National Center for Atmospheric Research (NCEP-NCAR I) reanalysis.

Reanalyses are derived from observations from multiple sources (including surface observations, aircraft, radiosondes, rocketsondes, and satellites) that are assimilated by global coupled land-atmosphere-ocean models to create spatially and temporally complete "observational" records. There are advantages and disadvantages of using reanalysis

products for this database, as opposed to individual measurement sources or various stratospheric analyses. These analyses include that from the Freie Universitat Berlin, which produces a database of continuous daily gridded synoptic-scale analyses based largely on radiosonde measurements, but only for three stratospheric levels for a 35 year period (Labitzke and



Collaborators, 2002), and from the NOAA Climate Prediction Center (CPC), which offers analyzed stratospheric temperatures at eight stratospheric levels based on satellite retrievals of the Advanced Microwave Sounding Unit (AMSU). The major advantage of reanalysis is to allow consideration of the evolution of SSWs and their impacts throughout the entire atmosphere with a spatial and temporal extent that is not feasible using individual measurements or stratospheric analyses

alone. A major disadvantage to using reanalysis is that due to sparse observations particularly in the pre-satellite era, stratospheric reanalysis is poorly constrained, especially above 10 hPa (Manney et al., 2003), and tropospheric reanalysis may be poorly constrained over oceans and remote regions (e.g., Bosilovich et al., 2008). Reanalyses can also suffer from upper-boundary effects and discontinuities due to model streams or changes in the observations being assimilated (Fujiwara et al., 2016; Labitzke and Kunze, 2005). These issues should not have a strong effect on the daily to seasonal timescales

documented in the SSWC, but should be kept in mind, especially for data above 10 hPa where the discontinuities are conspicuous.

Some biases and uncertainties in individual reanalysis products have been documented (see references in **Table 1**), and an evaluation of their stratospheric processes is currently the focus of an international effort by the Stratosphere-troposphere Processes And their Role in Climate (SPARC) Reanalysis Intercomparison Project (S-RIP; Fujiwara et al.,

2016). While initial studies have shown that stratospheric dynamics, variability, and coupling to the surface are reasonably simulated in reanalyses (Martineau and Son, 2010), particularly in the latest generation products (Martineau et al., 2016), the SSWC enables quick comparison between reanalyses of sudden stratospheric warming events and their evolution on daily timescales. This capability is important when considering the substantial volume of data needed to calculate the daily climatology and anomalies for each grid point and pressure level in each reanalysis.

Certain reanalysis output provided in the SSWC should be used with caution. For example, we provide the reanalysis ozone mass mixing ratio and total column ozone output (where available), as there are interesting changes in ozone following a SSW event (e.g., **Figure 3**). However, users should be aware that most reanalyses ozone fields are based on assimilated satellite measurements that utilize backscattered sunlight and cannot measure ozone during polar night. Reanalysis systems thus rely heavily on the model, which typically parameterizes heterogeneous chemistry, to simulate

ozone at high latitudes, leading to potentially high errors (Dethof and Hólm, 2004; Dragani, 2011).

The NOAA20CRv2c is unique among the reanalyses, because it assimilates only surface pressure observations. Thus the stratosphere is not constrained by any stratospheric observations, and the reanalysis winds are not realistic (Compo et al., 2011). However, because surface pressure observations do a reasonable job of constraining the model throughout the Northern Hemisphere troposphere (Compo et al., 2011), we include the NOAA20CRv2c to examine the tropospheric

impacts of SSWs, using SSW event dates given by the JRA-55 reanalysis (**Table 2**). The NOAA20CRv2c reanalysis provides the unique opportunity to examine tropospheric and stratospheric interaction prior to and following SSWs, when only the surface is constrained by observations.





### 2.2 Event Selection

*Major* SSWs occur when the wintertime polar stratospheric westerlies reverse to easterlies. In *minor warmings*, the polar temperature gradient reverses but the circulation does not, and in *final warmings*, the vortex breaks down but never recovers back to its climatological westerly state until the following boreal autumn. Because no unambiguous standard definition for major, minor, and final warmings yet exists (Butler et al., 2015), selecting SSW events to include in the Compendium is not straightforward.

The primary goal of the SSWC is to provide data for major SSWs, which have been found to have the largest surface impacts (Palmeiro et al., 2015). We recognize that any criteria we use may also select marginal events or miss events that perhaps should be considered major in terms of surface influences. We employ the following simple, commonly-used definition for major warmings (Charlton and Polvani 2007; hereafter CP07): the *central date* or *event date* of a SSW occurs when the daily-mean zonal-mean zonal winds at 10 hPa and 60° N first changes from westerly to easterly between November to March. The winds must return to westerly for 20 consecutive days between events. If the winds do not return to westerly for at least 10 consecutive days before 30 April, the warming is a final warming and is not included. The central dates for major NH SSWs in each reanalysis are provided in **Table 2**. We include in the SSW Compendium, for each reanalysis, every event detected in any reanalysis and shown in **Table 2** (for example, we include data for the 30 Nov 1958 event for all reanalyses extending back to 1958, even though it was only detected in NCEP-NCAR). This includes the NOAA20CRv2c, even though that reanalysis detects only a single event.

There are two main types of SSW: *displacement* events in which the stratospheric polar vortex is displaced off the pole; and *split* events in which the vortex splits into two or more vortices (**Figure 1**). Some SSWs are a combination of both types. There are a number of methods to determine the type of SSW. We do not attempt to classify event types here; however, we do provide the filtered (and unfiltered) absolute vorticity field at 10 hPa (see **Sect. 2.3**), which may enable classification of split-type SSWs according to the CP07 definition, in which the edges of the vortex are identified by the location of the maximum absolute vorticity gradient. We also provide potential vorticity (PV) interpolated onto isentropic surfaces, and geopotential heights at 10 hPa, both of which can be used to assess vortex moment diagnostics and determine the SSW type (Mitchell et al., 2011; Seviour et al., 2013; Waugh, 1997). We note that the vortex moment diagnostics detect some different dates of SSWs compared to CP07 (and these events are not included in the Compendium), but the provided data would allow classification of the included events.

While almost all SSWs occur in the NH, we did examine their occurrence in the SH in the reanalyses (**Table 3**). The relevant dates for zonal-mean zonal wind reversals at 10 hPa and 60° S were between July and October, and the winds must return to westerly for at least 10 consecutive days before 30 November. Keeping in mind that prior to 1979, there were hardly any observations of the SH polar stratosphere and so the reanalyses are highly unconstrained, the only event detected occurred in September 2002. This event is included in the SSWC.





### 2.3 Data Processing

The production flow chart for the SSWC is shown in **Figure 2**. We obtained the native horizontal and vertical pressure-level data for each reanalysis from various research data archives: NOAA20CRv2c and NCEP/NCAR I from the NOAA Earth System Research Laboratory/Physical Sciences Division (http://www.esrl.noaa.gov/psd/data/gridded/); JRA-

55, ERA-interim, and ERA-40 from the University Corporation for Atmospheric Research (UCAR) Research Data Archive (http://rda.ucar.edu/); and MERRA-2 from the Modeling and Assimilation Data and Information Services Center (MDISC, http://disc.sci.gsfc.nasa.gov/mdisc/).

We extracted the following fields (when available) on provided pressure-levels: zonal winds, meridional winds, temperatures, geopotential heights, Ertel's potential vorticity (PV), total column ozone, and ozone mixing ratio; and at the

surface: mean daily temperature, minimum daily temperature, maximum daily temperature, mean sea-level pressure, surface pressure, total precipitation liquid water equivalent, and total snowfall liquid water equivalent. Most raw reanalysis output is available every 6 hours (for pressure-level fields) and sometimes up to every 3 hours (for surface-level fields), but we computed daily-means of all fields for the SSWC. We interpolated pressure-level fields onto a 2.5° x 2.5° latitude-longitude grid, while the surface-level fields are maintained at native horizontal resolution. We retained data on provided pressure

levels, but we interpolated certain fields (PV and ozone mixing ratio) onto isentropic surfaces. Unless isentropic-level data is provided, we calculated potential temperature (θ) from temperature data on pressure levels using Eq. (1):

$$\theta = T(\frac{p_0}{p})^{R/C_p} \quad , \tag{1}$$

where $T$ and $p$ are atmospheric temperature and pressure respectively, $p_0$ is a reference pressure defined as 1000 hPa, $R$ is the molar gas constant, and $c_p$ is the specific heat capacity at constant pressure. The data, either on pressure or isentropic levels,

are interpolated at each time step onto 10 common isentropes (330, 350, 400, 450, 500, 550, 600, 700, 850, and 1000 K). Note that in JRA-55, isentropic-level data are provided but not at the 1000 K surface, so in the SSWC missing values are indicated for this theta level.

There are two types of output provided by the SSWC: climatological statistics and event-based data. Climatological statistic files include the mean and standard deviations of all output fields, and percentiles from the climatological

distribution for a selection of surface fields: minimum and maximum surface temperature and precipitation. The climatological statistics are defined at each spatial point for 366 days spanning 01 July – 30 June. The climatological mean is based on the entire time period of each reanalysis (**Table 1**). To calculate the climatological mean, we first calculate the mean of each day of the year over the full record. Then we calculate the Fourier transform of this daily mean climatology and retain the first four harmonics of the Fourier series (e.g., Wilks, 2006). This methodology smooths out the raw daily

climatology while preserving low-frequency variability. The standard deviation is then calculated by taking the square root of the squared deviations in the raw daily data from this smoothed climatological mean. Percentiles are calculated following



a method described in Zhang et al., (2005; c.f. equation (1)). Chosen percentiles are 5, 10, 90, and 95%. These statistics are calculated using the entire data record.

Event-based files contain full field, anomaly, and derived fields for the 60 days prior to and following each SSW event in **Tables 2 and 3**. Anomalies are calculated using the smoothed climatology for each field, using the entire data

record for each reanalysis. We caution that, while the climatologies for different time periods are generally quite similar, using different periods for the climatology for each reanalysis means that differences in reanalysis anomaly fields may partially be a result of the climatology chosen. In addition to full fields and anomalies, we derive a number of useful diagnostics for understanding dynamic processes and surface climate surrounding SSW events, as described below:

(1) The maximum and minimum daily temperatures (NCEP-NCAR I provides this output; we calculate these values

for the other reanalyses).

(2) Standardized geopotential height anomalies. The geopotential heights are standardized by subtracting the mean and dividing by the standard deviation for the particular day of year and gridpoint.

(3) Absolute vorticity ($\omega_a$) at 10 hPa. This is calculated from the 2.5° x 2.5° gridded zonal and meridional wind fields, using the vorticity equation in spherical coordinates:

$$\omega_a = \zeta + f = (\frac{1}{a}\frac{\partial v}{\partial \lambda} - \frac{1}{a\cos\phi}\frac{\partial(u\cos\phi)}{\partial\phi}) + f \ , \qquad (2)$$

where $\zeta$ is relative vorticity (defined by the parenthetical terms on the right-most side of the equation), $f$ is the Coriolis term ($2\Omega sin\phi$), $a$ is the Earth's radius, $\phi$ is the latitude in radians, $\lambda$ is the longitude in radians, $u$ is the zonal wind, and $v$ is the meridional wind.

(4) Filtered absolute vorticity at 10 hPa. Here the absolute vorticity has been subject to a spherical smoothing

procedure, in which the absolute vorticity is transformed to spherical harmonic space and subsequently transformed back while retaining only the first 11 harmonic coefficients. This filtering is part of CP07's event type determination algorithm.

(5) Zonal-mean eddy meridional heat flux ($v'T'$), and its wavenumber 1 and 2 components, as a function of pressure level and latitude. Here the primes (') indicate deviations from the zonal mean. These are calculated using daily data. The wavenumber components are found by applying a Fourier transform to the longitude dimension.

(6) Zonal-mean eddy meridional momentum flux ($u'v'$), and its wavenumber 1 and 2 components, as a function of pressure level and latitude.

(7) The Northern Annular Mode (NAM) and the Southern Annular Mode (SAM) indices. The NAM/SAM patterns are calculated as the first empirical orthogonal function (EOF) of daily-mean, zonal-mean geopotential height anomalies from 20-90°N/S. The NAM/SAM indices are the principal component time series corresponding to the first EOF for each

hemisphere (Baldwin and Thompson, 2009). In the stratosphere, the annular mode is related to the strength of the polar vortex; in the troposphere, the annular mode is related to shifts in the tropospheric storm tracks (Gerber et al., 2012; Thompson et al., 2000).





(8) Extreme events. For each grid space, either a 0 or 1 is given if the daily precipitation, minimum temperature, or maximum temperature anomaly exceeds a certain threshold. For precipitation, the anomaly must exceed the 95[th] percentile. Temperature anomalies must either be less than the 5[th] or 10[th] percentile, or greater than the 90[th] or 95[th] percentile.

(9) Time series of the location of maximum stratospheric warming, within the region of 30-90° latitude and between 300 hPa to 1 hPa (or as high as the reanalysis provides). This includes the geopotential height, latitude, longitude, and pressure of the maximum temperature anomaly. Time series of the location of the minimum zonal wind anomaly are also included for the same region.

(10) Time from the SSW event at which the zonal-mean zonal wind becomes easterly, as a function of pressure and latitude.

(11) Pressure level at which the zonal-mean zonal wind becomes easterly, as a function of time and latitude.

Finally, a number of climate indices based on independent observations (not reanalysis data) have been included to provide a sense of other sources of climate variability that may be contributing to both the forcing of individual SSWs and the surface climate impacts. These include:

(1) Measures of the phase of the El Niño-Southern Oscillation (ENSO). These indices allow the user to assess the state of the tropical Pacific, which has important wintertime effects on mid-latitude climate. SSWs have been found to occur in 80% of El Niño winters (Butler and Polvani, 2011), and may modify the El Niño teleconnections when they occur (Butler et al., 2014; Richter et al., 2015). The Multivariate ENSO Index (MEI) is calculated as the first principal component of six different observed variables combined. The MEI data is from NOAA Physical Sciences Division (PSD): http://www.esrl.noaa.gov/psd/enso/mei/table.html. In addition to the MEI, we also provide the Oceanic Niño Index (ONI) and the Southern Oscillation Index (SOI). The ONI is calculated as the 3-month running mean of sea surface temperature anomalies in the Niño 3.4 region, based on a centered 30-year base period updated every five years. The ONI data is from the NOAA CPC: http://www.cpc.ncep.noaa.gov/products/analysis_monitoring/ensostuff/detrend.nino34.ascii.txt. The SOI is calculated as the difference between the standardized sea level pressure at Tahiti and Darwin. The SOI data is from the NOAA CPC: http://www.cpc.ncep.noaa.gov/data/indices/soi. All of these indices have been interpolated from monthly data to daily data.

(2) The Outgoing long-wave radiation Madden Julian Oscillation (MJO) Index (OMI) amplitude and phase. SSWs may be related to the anomalous convection generated by the MJO during certain phases (e.g., Garfinkel et al., 2014). The OMI daily data is from NOAA PSD: http://www.esrl.noaa.gov/psd/mjo/mjoindex/omi.1x.txt.

(3) The equatorial zonal winds measured by radiosondes near the equator, provided at 10, 30, 50, and 70 hPa, as a measure of the Quasi-Biennial Oscillation (QBO). The QBO is thought to modulate the frequency of SSWs via changes in wave propagation (Baldwin et al., 2001; Dunkerton et al., 1988), perhaps in relation to the solar cycle (Labitzke et al., 2006). The QBO data is provided by Freie Universitat of Berlin: http://www.geo.fu-berlin.de/en/met/ag/strat/produkte/qbo/. These have been interpolated from monthly data to daily data.





We acknowledge that other variables and indices may be useful for examining SSW dynamics, such as the Eliassen-Palm flux vector components or Transformed-Eulerian Mean diagnostics. Some of these diagnostics could be calculated using the provided daily data on pressure levels, though this may be imprecise relative to calculations on native model levels. Model level data is often used for analyzing transport and processes near the tropopause, where vertical resolution on
provided pressure levels may be inadequate or may introduce interpolation errors. Regardless, the SSWC is useful for a wide range of applications, as featured in the next section.

## 3 Applications

Here we highlight three types of potential applications of the SSWC: (i) composite analysis, (ii) individual event analysis, and (iii) reanalysis intercomparison.

### 3.1 Composite analysis

Assessing the composite response to SSWs is useful for separating the signals from internal noise, and identifying where the signal is robust. **Figure 3** shows, as a function of pressure level and time before and after the event, (a) zonal-mean zonal winds at 60° N and zonal-mean temperature anomalies averaged from 50-90° N, (b) the Northern Annular Mode index at each pressure level, and (c) ozone mixing ratios from 60-90° N, composited over all 41 Northern Hemisphere SSW
events (**Table 2**), using the JRA-55 reanalysis. **Figure 4** shows the surface response composited over the 60 days following the central date of all SSWs, including (a) mean sea level pressure anomalies, (b) surface temperature anomalies, and (c) precipitation anomalies.

These two figures illustrate several important and well-known features of SSWs and their impacts on circulation and surface climate (e.g., Baldwin and Dunkerton, 2001). In the stratosphere, the zonal-mean zonal winds change from
westerly to easterly at 10 hPa and 60° N at lag zero (the central date), as constructed by the SSW definition (**Fig 3a**). The zonal wind reversal is strongest near ~3 hPa. In the composite, a complete wind reversal extends from 1 hPa down to ~10 hPa, but a deceleration of the zonal winds extends throughout the whole stratosphere. The peak warming of the stratosphere occurs ~1 day before the peak zonal wind reversal, and its location at ~7 hPa is consistent with peak zonal wind decreases at higher altitudes, per the thermal wind relationship. At 10 hPa and higher, the zonal winds and temperatures rebound quickly
after the SSW, reforming a colder westerly vortex above 10 hPa after 10-15 days. In the lower stratosphere, warmer, weaker vortex conditions persist 60 days following the SSW due to slow radiative time scales (Newman and Rosenfield, 1997). These changes near the tropopause may increase the persistence of the negative NAM phase in the troposphere (**Fig 3b**), potentially providing a source of predictive skill for up to 60 days after the occurrence of the SSW (Maycock and Hitchcock, 2015). Following the SSW, the stratospheric ozone over the polar cap is greatly enhanced (**Fig 3c**), both due to the increased
transport of ozone-rich air into the stratosphere via the residual mean circulation, and the horizontal mixing of high-ozone air



into the region as the low-ozone region of the polar vortex is moved off the pole (either in one or two lobes, depending on whether a split- or displacement- type event has occurred).

At the surface, the composite response in mean sea level pressure anomalies comprises an anomalous high over the polar cap and Greenland, and an anomalous low over the North Atlantic; a pattern which projects well onto the negative

phase of the NAO, the regional equivalent of the NAM (**Fig 4a**). The associated surface temperature anomalies include significant warming over western Greenland and eastern Canada, and strong cold air outbreaks over much of northern Europe, Asia, and the eastern United States (**Fig 4b**). Conditions are also anomalously wet over western and central Europe, and dry over Scandinavia (**Fig 4c**).

Composite analysis could also be used to consider differences in SSW evolution and impacts in relation to other

factors, such as the differences between split- and displacement- type events, or differences between events that occur in El Niño or La Niña winters or different phases of the MJO. **Figure 5** highlights the differences in the evolution of the 500 hPa geopotential height anomalies prior to and after a SSW, during La Niña versus El Niño winters. Here we use the December-January-February ONI index to classify El Niño and La Niña years, with winters with ONI exceeding +0.5° C defined as El Niño years, and winters with ONI below -0.5° C defined as La Niña years. While the sample size for these composites is

small (13 events during El Niño years, 9 events during La Niña years), some major features are apparent; for example, the trough during El Niño and the ridge during La Niña in the North Pacific are evident throughout the evolution of the SSW. Note, however, the intensification of low pressure anomalies in the northwest Pacific in the 60 days prior to SSWs in both El Niño and La Niña winters, a feature theorized in Garfinkel et al. (2012) to amplify planetary scale waves from the troposphere into the stratosphere and weaken the stratospheric polar vortex. During El Niño winters, the tropospheric

circulation pattern is strongest over North America in the days prior to a SSW, but strongest over the North Atlantic after a SSW. During La Niña winters, the anomalies over Greenland and Europe change sign before and after a SSW event, demonstrating the role of SSWs on wintertime climate over the North Atlantic-European region.

### 3.2 Individual event analysis

While compositing is useful for highlighting robust features of SSWs, the dynamic evolution and surface climate

anomalies before and after each individual SSW can vary widely. The SSWC can be used to demonstrate this range of variability. **Figure 6** illustrates the differences in the tropospheric climate following two similar split-type SSWs, one in January 1985 and the other in January 2009. In both events, the polar vortex split into two lobes: the one associated with the greatest warming anomalies centered over Canada, and the other centered over northern Europe and Asia (**Fig 6a, b**). The 2009 split SSW had a larger lobe that extended over most of Eurasia, but otherwise the stratospheric evolution was quite

similar.

However, the subsequent surface and tropospheric response in the weeks following the events differed in several ways. The 500 hPa height anomaly pattern following the 1985 event projects strongly onto the negative NAO pattern (**Fig 6c**), with positive height anomalies over Greenland and negative height anomalies over the North Atlantic. This pattern is





associated with much lower surface temperature anomalies over much of Europe and Asia. However, the height anomalies in the two months following the 2009 split-type event do not look like the negative NAO phase, though there are weakly positive height anomalies over the Arctic and two centers of low height anomalies over Europe and Asia (**Fig 6d**). Temperature advection associated with these anomalous low pressure centers may explain the regional cold air experienced

over Asia and central Europe. Comparison of these two events shows how different modes of climate variability can impact the tropospheric climate during the period after a substantial SSW event. While 1985 and 2009 were both (essentially) La Niña winters (2009 misses official La Niña classification by the NOAA Climate Prediction Center by 0.1° C), the location and strength of the North Pacific ridge during these two years was quite different. Other aspects of climate variability, such as the QBO, sea ice, or the MJO, may have been playing a role in the tropospheric climate during these time periods.

The SSWC allows easy evaluation of the spread among individual events for different features of SSWs. **Figure 7** shows time series of the (a) amplitude and (b) latitude of the maximum temperature anomaly (that occurs within the range of 30-90° latitude and 300 hPa to 1 hPa), and (c) the 200 hPa 40-70° N eddy heat flux anomaly. On average, the maximum temperature anomaly of ~50 K peaks 1-2 days prior to the zonal wind reversal (**Fig 7a**; bold black line), but the amplitude and timing varies substantially among the individual events (colored lines), with values from 10 to almost 100 K. Likewise,

the mean latitude where the temperature maximizes tends to fall between 60-70° N (**Fig 7b**), but ranges from ~45° N to the pole. The 200 hPa heat flux anomaly represents the incoming heat fluxes from the troposphere via vertically propagating waves, which amplify and peak prior to the SSW (Polvani and Waugh, 2004; Sjoberg and Birner, 2014); but during any individual year, there may be pulses of large heat fluxes that do not result in a SSW (**Fig 7c**).

**3.3 Reanalysis intercomparison**

Finally, the SSWC includes data from 6 different reanalyses, both to aid in reanalysis intercomparison projects such as S-RIP and to allow users the ability to assess the robustness of SSW features in different products. **Figure 8** demonstrates how these differences manifest during the Jan 2013 SSW event, for (a) a modern reanalysis product (MERRA2), (b) an older reanalysis product with low model top (NCEP1), and (c) a reanalysis that only assimilates observations at the surface and has a strong bias in the stratosphere (NOAA20CR). In MERRA2, there is strong weakening of the zonal wind anomalies at 60°

N, which starts near 1 hPa around the event date and descends over time to the tropopause (**Fig 8a, left panel**). These anomalies are also evident in NCEP1, but output is only available up to 10 hPa, and the anomalies at 10 hPa tend to be slightly smaller than those in MERRA2 (**Fig 8b**). The NOAA20CRv2c makes an interesting comparison, because the model stratospheric winds are too strong but the surface is constrained by assimilated observations (**Fig 8c**). This means that although NOAA20CRv2c does not capture the SSW event, the surface and tropospheric response contains information about

the impact of this stratospheric event. On the other hand, the mid- to upper-tropospheric zonal wind anomalies after the SSW event in NOAA20CRv2c are smaller (more positive) than in either NCEP1 or MERRA2, suggesting that the lack of stratospheric processes limits the ability of this reanalysis to capture the tropospheric climate response following major breakdowns of the polar vortex.



The surface temperature anomalies and the 200 hPa geopotential height anomalies for days 30-60 after the 2013 SSW are shown in the right hand panels of **Fig 8**. In the SSWC, surface variables are provided at their native horizontal resolution, which is reflected in these panels in the surface temperature anomalies. MERRA2 has the highest horizontal resolution, so more regional structure and detail are apparent. The cold anomalies over Asia and parts of the Arctic, and the

tropospheric circulation anomalies at 200 hPa (particularly in regions impacted by stratosphere-troposphere coupling, such as the North Atlantic), are weaker in the NOAA20CR relative to MERRA2 and NCEP1. Regional differences between all three reanalyses can be seen, particularly in the polar cap region where observations may not be available to constrain the reanalysis system.

**4 Data Usage and Availability**

The SSWC is designed to be a public domain product that allows the user either to use the data as packaged or to step into the production process and re-generate parts of the database with customized configurations. A flow chart of these options is shown in **Figure 2**. For example, if the user would like to use a different set of event dates or a different climatology, they may use the provided code and documentation to extract full fields from their reanalysis product of choice and to generate new anomaly and derived fields. Nonetheless, one major advantage of the SSWC is that both the full fields

and anomalies are provided (as well as the climatology), so that users can avoid downloading the terabytes of data needed to calculate the daily climatology and anomaly fields.

The SSW Compendium has been archived at NOAA's NCEI (doi: http://dx.doi.org/10.7289/V5NS0RWP) in CF-compliant netCDF-4 format. The data are compressed using short integer (16-bit) packing, resulting in a full size of 300 GB for the SSWC. Some, but not all, programming platforms will properly read packed data and account for missing values.

Care must be taken while reading packed data, or missing values may be unknowingly counted as finite data points.

A User's Guide to the SSWC dataset is provided to describe the included variables and the file format. A Production Guide and source code in IDL format are provided in case a user would like to re-create their own version of the SSWC. We anticipate future updates to the Compendium when new SSWs occur, for those reanalysis products that proceed operationally in the future. When the Compendium is updated with a new SSW event, the climatologies and anomalies for

all events will be updated, based on the full period of the new record. When publishing results based on the SSWC, users should clearly state what version/climatology is being used, in order to allow reproducible results. A subset of the SSWC can be downloaded, plotted, or animated at: http://www.esrl.noaa.gov/csd/groups/csd8/sswcompendium/.

The ability to readily perform (i) composite analysis, (ii) individual event analysis, and (iii) reanalysis intercomparison are one of the main goals of the SSW Compendium. The SSWC will hopefully allow users to highlight the

role of stratosphere-troposphere processes and the importance of major SSW events in wintertime climate, and will provide a comprehensive database to compare with and improve model simulation of these events.



## 5 Summary

The SSWC database provides a simple and computationally inexpensive way to generate, download, and plot information on historical SSW events and their evolution and impacts on daily timescales. The database is designed to be used "as is", but the end-user also has the ability to use the source code to customize the database to meet their specific needs. The inclusion of six different reanalysis products and a set of full, anomaly, and derived fields for every major SSW in the historical record allows several different applications of the SSWC. The ability to readily perform (i) composite analysis, (ii) individual event analysis, and (iii) reanalysis intercomparison for projects such as S-RIP will hopefully allow users to highlight the role of stratosphere-troposphere processes and the importance of major SSW events in wintertime climate, and will provide a comprehensive database to compare with and improve model simulation of these events.

**Acknowledgements.** This work was funded by the NOAA Climate Program Office.

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



**Table 1: The reanalyses included in the SSW Compendium**.

| Reanalysis | Time Period | Reference | Native horizontal resolution | Vertical Resolution (model/pressure levels) | Model Top |
|---|---|---|---|---|---|
| ERA-40 | 1958-2002 | (Uppala et al., 2005) | 1.125°x1.125° | 60/23 | 0.1 hPa |
| ERA-interim | 1979-2014 | (Dee et al., 2011) | 0.75°x0.75° | 60/23 | 0.1 hPa |
| JRA-55 | 1958-2014 | (Kobayashi et al., 2015) | 1.25°x1.25° | 60/37 | 0.1 hPa |
| MERRA2 | 1980-2014 | (Molod et al., 2015) | 0.5°x0.667° | 72/42 | 0.01 hPa |
| NCEP-NCAR I | 1958-2014 | (Kalnay et al., 1996) | 2.5°x2.5° | 28/17 | 3 hPa |
| NOAA20CRv2c | 1958-2014 | (Compo et al., 2011) | 2°x2° | 28/24 | 10 hPa |





**Table 2: The central dates of NH SSWs detected in each reanalysis product[a].**

|    | ERA-40 | ERA-interim | JRA-55 | MERRA2 | NCEP-NCAR I | NOAA20CR |
|----|--------|-------------|--------|--------|-------------|----------|
| 1  | 31-Jan-58 |          | 30-Jan-58 |          | 30-Jan-58 | **** |
| 2  | ****      |          | ****      |          | 30-Nov-58 | **** |
| 3  | 17-Jan-60 |          | 17-Jan-60 |          | 16-Jan-60 | **** |
| 4  | 28-Jan-63 |          | 30-Jan-63 |          | ****      | **** |
| 5  | ****      |          | ****      |          | 23-Mar-65 | **** |
| 6  | 16-Dec-65 |          | 18-Dec-65 |          | 8-Dec-65  | **** |
| 7  | 23-Feb-66 |          | 23-Feb-66 |          | 24-Feb-66 | **** |
| 8  | 7-Jan-68  |          | 7-Jan-68  |          | ****      | **** |
| 9  | 28-Nov-68 |          | 29-Nov-68 |          | 27-Nov-68 | **** |
| 10 | 13-Mar-69 |          | ****      |          | 13-Mar-69 | **** |
| 11 | 2-Jan-70  |          | 2-Jan-70  |          | 2-Jan-70  | **** |
| 12 | 18-Jan-71 |          | 18-Jan-71 |          | 17-Jan-71 | **** |
| 13 | 20-Mar-71 |          | 20-Mar-71 |          | 20-Mar-71 | **** |
| 14 | 31-Jan-73 |          | 31-Jan-73 |          | 2-Feb-73  | **** |
| 15 | 9-Jan-77  |          | 9-Jan-77  |          | ****      | **** |
| 16 | 22-Feb-79 | 22-Feb-79 | 22-Feb-79 |          | 22-Feb-79 | **** |
| 17 | 29-Feb-80 | 29-Feb-80 | 29-Feb-80 | 29-Feb-80 | 29-Feb-80 | 18-Mar-80 |
| 18 | ****      | ****      | 6-Feb-81  | ****     | ****      | **** |
| 19 | 4-Mar-81  | 4-Mar-81  | 4-Mar-81  | 4-Mar-81 | ****      | **** |
| 20 | 4-Dec-81  | 4-Dec-81  | 4-Dec-81  | 4-Dec-81 | 4-Dec-81  | **** |
| 21 | 24-Feb-84 | 24-Feb-84 | 24-Feb-84 | 24-Feb-84 | 24-Feb-84 | **** |
| 22 | 1-Jan-85  | 1-Jan-85  | 1-Jan-85  | 1-Jan-85 | 2-Jan-85  | **** |
| 23 | 23-Jan-87 | 23-Jan-87 | 23-Jan-87 | 23-Jan87 | 23-Jan-87 | **** |
| 24 | 8-Dec-87  | 8-Dec-87  | 8-Dec-87  | 8-Dec-87 | 8-Dec-87  | **** |
| 25 | 14-Mar-88 | 14-Mar-88 | 14-Mar-88 | 14-Mar-88 | 14-Mar-88 | **** |
| 26 | 21-Feb-89 | 21-Feb-89 | 21-Feb-89 | 21-Feb-89 | 22-Feb-89 | **** |
| 27 | 15-Dec-98 | 15-Dec-98 | 15-Dec-98 | 15-Dec-98 | 15-Dec-98 | **** |
| 28 | 26-Feb-99 | 26-Feb-99 | 26-Feb-99 | 26-Feb-99 | 25-Feb-99 | **** |
| 29 | 20-Mar-00 | 20-Mar-00 | 20-Mar-00 | 20-Mar-00 | 20-Mar-00 | **** |
| 30 | 11-Feb-01 | 11-Feb-01 | 11-Feb-01 | 11-Feb-01 | 11-Feb-01 | **** |
| 31 | 31-Dec-01 | 30-Dec-01 | 31-Dec-01 | 30-Dec-01 | 2-Jan-02  | **** |
| 32 | 18-Feb-02 | ****      | ****      | ****     | ****      | **** |
| 33 |           | 18-Jan-03 | 18-Jan-03 | 18-Jan-03 | 18-Jan-03 | **** |
| 34 |           | 5-Jan-04  | 5-Jan-04  | 5-Jan-04 | 7-Jan-04  | **** |
| 35 |           | 21-Jan-06 | 21-Jan-06 | 21-Jan-06 | 21-Jan-06 | **** |
| 36 |           | 24-Feb-07 | 24-Feb-07 | 24-Feb-07 | 24-Feb-07 | **** |
| 37 |           | 22-Feb-08 | 22-Feb-08 | 22-Feb-08 | 22-Feb-08 | **** |
| 38 |           | 24-Jan-09 | 24-Jan-09 | 24-Jan-09 | 24-Jan-09 | **** |
| 39 |           | 9-Feb-10  | 9-Feb-10  | 9-Feb-10 | 9-Feb-10  | **** |
| 40 |           | 24-Mar-10 | 24-Mar-10 | 24-Mar-10 | 24-Mar-10 | **** |
| 41 |           | 06-Jan-13 | 07-Jan-13 | 07-Jan-13 | 07-Jan-13 | **** |

[a]These are the detected events in each reanalysis, but in the SSWC we provide data for all dates shown in this Table for all reanalyses.



**Table 3: The central dates of the SH SSW detected in each reanalysis product**.

| | ERA-40 | ERA-interim | JRA-55 | MERRA2 | NCEP-NCAR I | NOAA20CR |
|---|---|---|---|---|---|---|
| 1 | | 25-Sep-02 | 26-Sep-02 | 26-Sep-02 | 26-Sep-02 | **** |



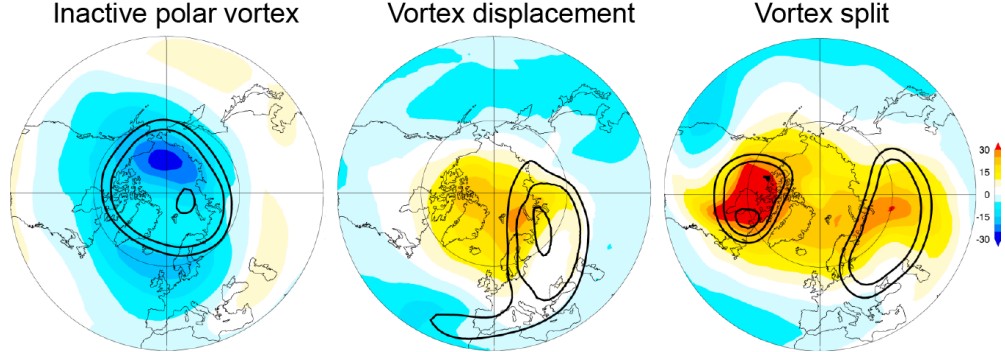

**Figure 1:** Temperature anomalies at 10 hPa (shading, [K]) and the potential vorticity at 550 K [contours shown for 75, 100, and 125 PV units] during (left) an "inactive" (or strong) phase of the polar vortex (~ 9 Jan 2009), (center) a vortex displacement following the 23 Jan 1987 event, and (right) a vortex split following the 24 Jan 2009 event. MERRA2 reanalysis is used.



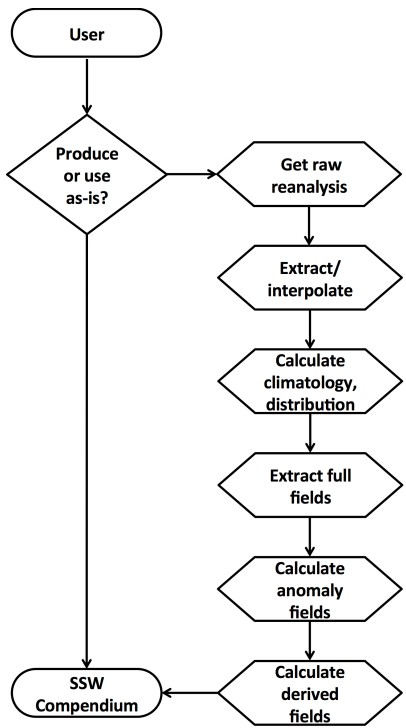

**Figure 2: Flow chart showing how the SSWC can be used "as is" or the different steps to produce the dataset.**



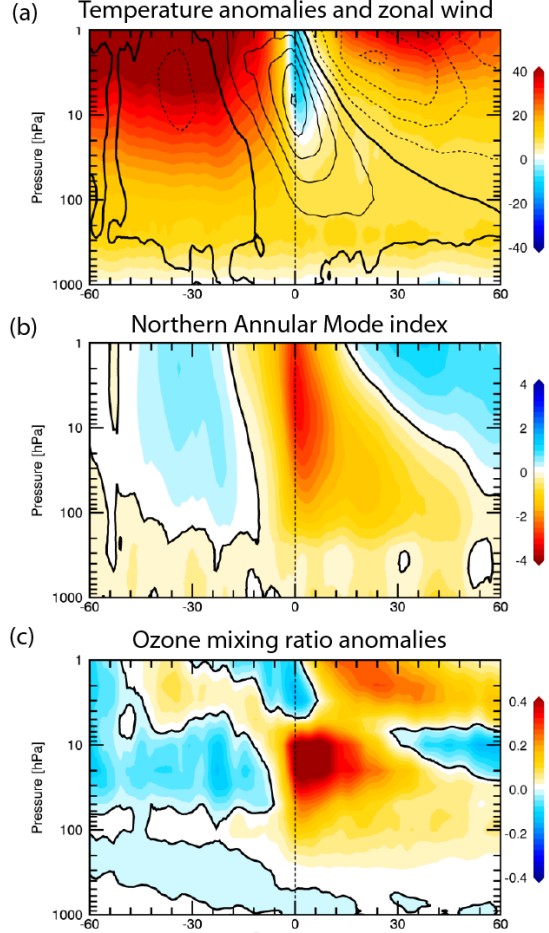

**Figure 3: Composites of the 60 days before and after historical SSWs in the JRA-55 reanalysis for (a) temperature anomalies averaged from 50-90° N (contours, [K]) and zonal-mean zonal winds at 60° N (shading, [m s⁻¹]); (b) the Northern Annular Mode index (NAM) [stdevs]; and (c) ozone mass mixing ratio anomalies from 60-90° N [ppmv].**





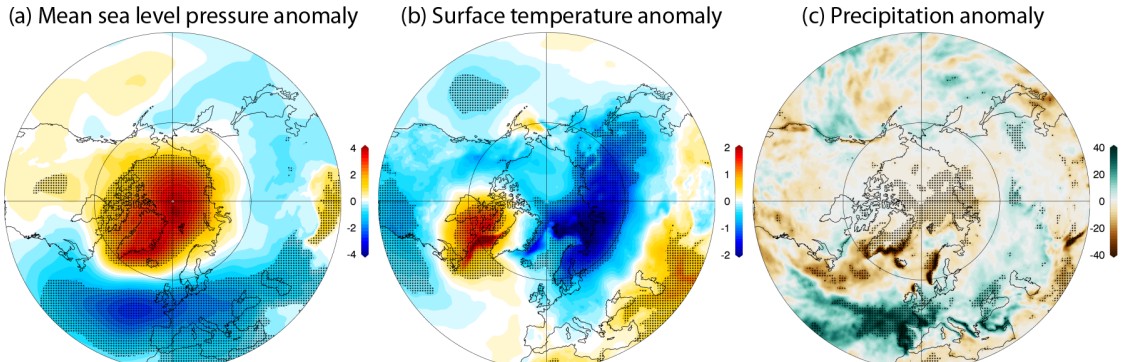

**Figure 4: Composites of the 60 days after historical SSWs in the JRA-55 reanalysis for (a) mean sea level pressure anomalies [hPa]; (b) surface temperature anomalies [K]; and (c) precipitation anomalies [mm]. The stippling indicates regions that are significantly different at the 95% level from the climatology.**




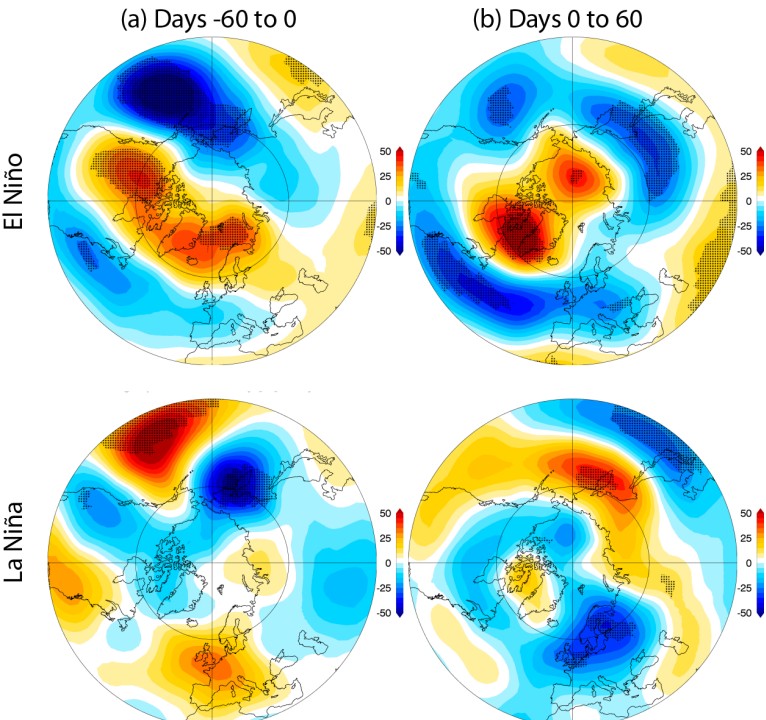

**Figure 5: Composites of the 500 hPa geopotential height anomalies [m] in JRA-55 reanalysis for (a) days -60 to 0 prior to historical SSWs, and (b) days 0 to +60 after historical SSWs, for (top row) El Niño winters and (bottom row) La Niña winters. The stippling indicates regions that are significantly different at the 95% level from the climatology. There are 13 events during El Niño winters and 9 events during La Niña winters. Here, if two SSWs occurred in one winter, we only considered the first event of the winter to avoid oversampling.**




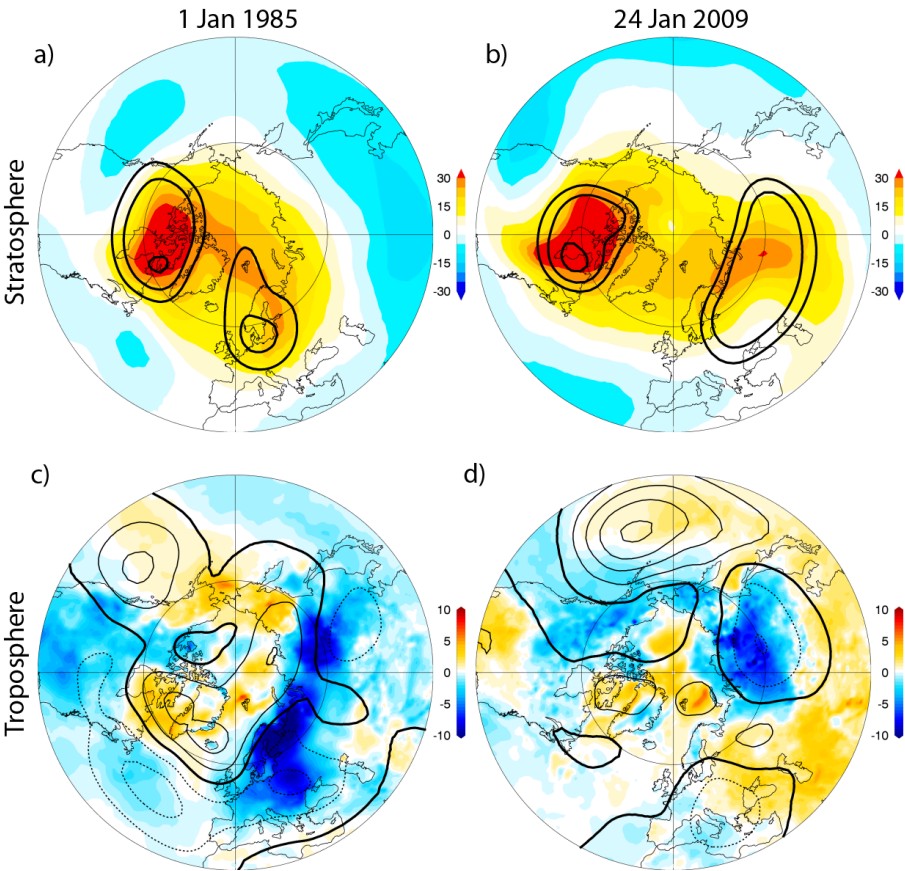

**Figure 6: Comparison of two split-type SSW events, (a,c) 1 Jan 1985 and (b,d) 24 Jan 2009, for ERA-interim reanalysis. The top row (a,b) shows the 10 hPa temperature anomalies (shading, [K]) and the potential vorticity at 550 K [contours shown for 75, 100, and 125 PV units] at +4 days after the central date of the event. The bottom row (c,d) shows the surface temperature anomalies (shading, [K]) and the 500 hPa geopotential height anomalies [contour interval is 50 m; zero line is bold] averaged days 0-60 after the central date of the event.**




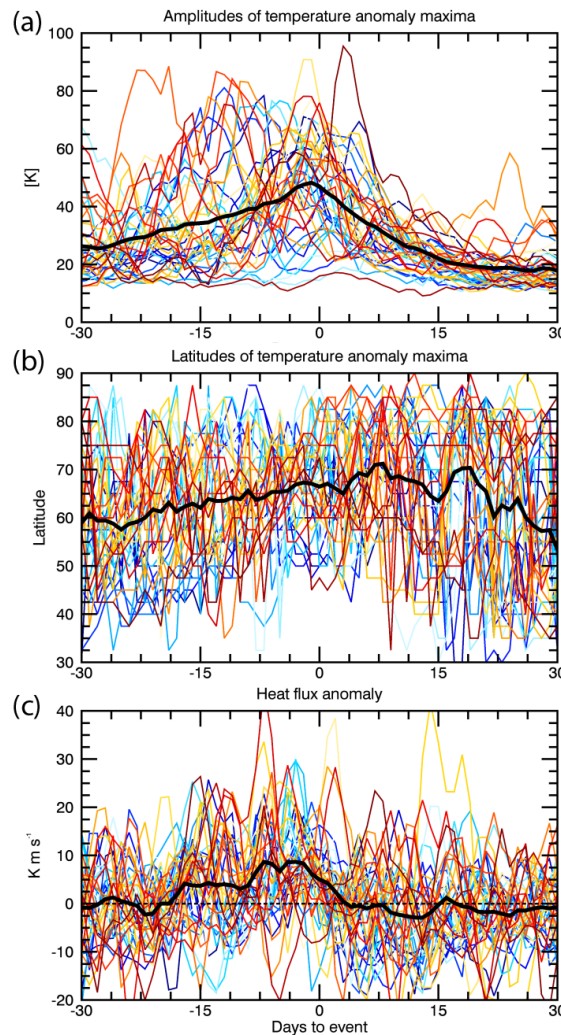

**Figure 7: Time series for the 30 days prior to and after the event date of major SSWs in the JRA-55 reanalysis, of (a) the amplitude of the maximum temperature anomaly (within the region 30-90° latitude and 300 hPa to 1 hPa; [K]), (b) the latitude of the maximum temperature anomaly within that same region [degrees latitude], and (c) the anomalous eddy heat flux [K m s$^{-1}$] at 200 hPa.**

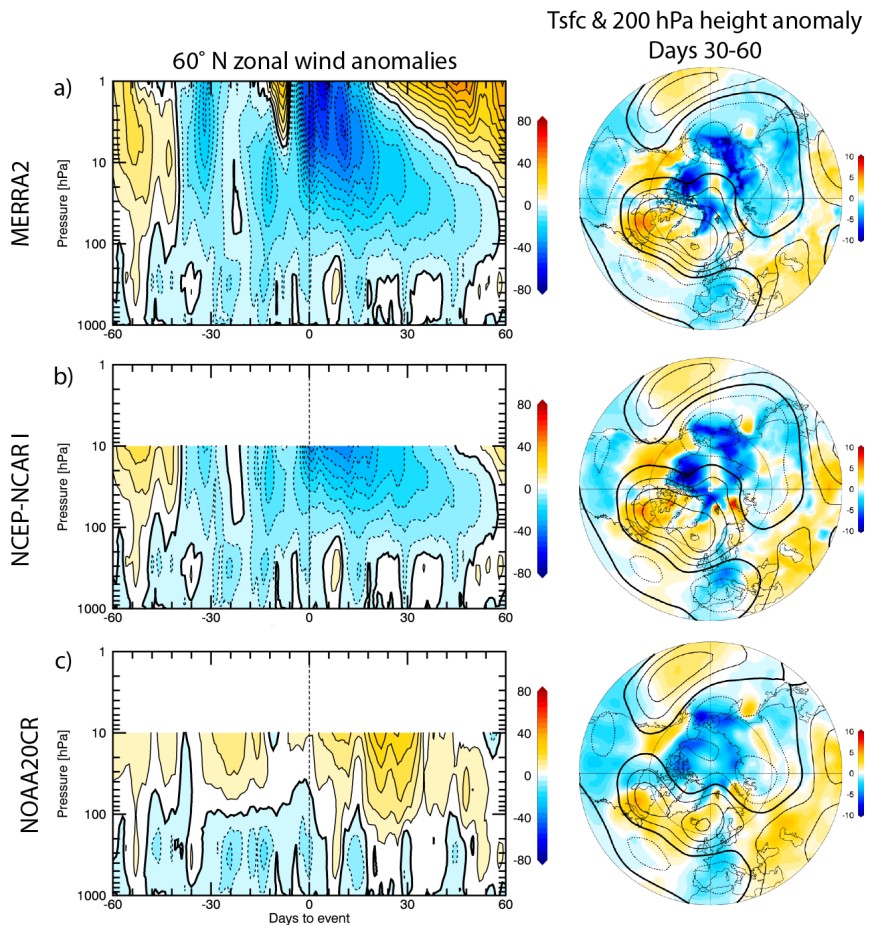

**Figure 8: Comparison of three different reanalysis products for the 7 Jan 2013 SSW event: (a) MERRA2, (b) NCEP-NCAR I, and (c) NOAA20CR. The left column shows 60° N zonal-mean zonal wind anomalies [m s⁻¹] as a function of time from the central date and pressure level. The right column shows the surface temperature anomalies (shading, [K]) and 200 hPa geopotential height anomalies [contour interval is 50 m] averaged over days 30-60 following the central date.**