# Peer review of "A Sudden Stratospheric Warming Compendium"

_Earth System Science Data, 2016_

## Referee Comment (RC1) · W. Seviour (Referee) · 20 Oct 2016

Summary:

The authors describe the Sudden Stratospheric Warming Compendium (SSWC) data set; a collection of data relevant for studying the dynamics and impacts of SSWs from 1958-2014. The SSWC consists of data from 6 reanalysis products: ERA-40, ERA-Interim, JRA-55, MERRA2, NCEP-NCAR I, and NOAA20CRv2c. Pressure-level data, including winds, temperature, geopotential height, PV and ozone mixing ratio is available, along with surface temperature, pressure, and precipitation. These fields can be downloaded as either climatological values, full fields, or anomalies over the 60 days before and after each SSW event. El-Nino-Southern Oscillation, Madden Julian Oscillation, and Quasi-Biennial Oscillation indices are also provided, since these are known to have links with SSWs and their impacts. The authors provide examples of how the SSWC may be used to study composites of SSW events, individual events, as well as

to compare reanalyses. The data set is provided in CF-compliant netCDF-4 format, and hosted publicly at NOAA's National Centers for Environmental Information (NCEI).

This is a well-written paper which I have enjoyed reading, and I hope (and expect) the SSWC data set will a valuable resource for the community. The data is easily accessible, and I applaud the authors for using the widely accepted CF metadata conventions. I recommend this paper for publication, and have included some minor comments below, which I hope the authors find constructive.

Comments:

Data Accessibility:

The drop-down menu for downloading data at http://esrl.noaa.gov/csd/groups/csd8/sswcompendium/ appears not to be working, and only lists JRA-55 on the menu for reanalysis products. While data can be downloaded from the NCEI at https://www.ncei.noaa.gov/data/ssw/, the file names are quite cryptic. I would encourage the authors to ensure that the drop-down menu download is working, and include all of the fields in the SSWC. This would improve the accessibility of SSWC data, particularly for new users.

Text:

1. Line 4/Page 2: Make clear the sign of the impact of SSWs on cold air outbreaks (i.e. "..., such as increasing the likelihood of cold air outbreaks ...")

2. L21/P2: Change "40 degrees K" to "40 K", since Kelvin is absolute, not a 'degree'.

3. L4/P5: "... in final warmings, the vortex breaks down but never recovers back to its climatological westerly state until the following boreal autumn": The climatological state following a final warming may actually be easterly (if it is a late final warming). I think saying "in final warmings, the vortex breaks down and becomes easterly until the following boreal autumn" is more accurate.

4. L12/P5: Perhaps mention the motivation for choosing 20 days between events.

Radiative time scales?

5. L9/P6: Total column ozone is not on pressure-levels as this implies, but is vertically integrated.

6. L19/P6: Mention the value of c_p used for calculating isentropic levels.

7. L20/P6: Could a bit more information on the interpolation be provided? Was it linear interpolation?

8. L9/P7: For data sets that do not provide daily maximum/minimum temperature, how accurate is the calculation from 6-hourly or 3-hourly data? Are the minimum/maximum 6/3-hourly values used, or is there interpolation between them to calculate these values?

9. L24/P8 and L34/P8: How are monthly-mean values interpolated to give daily values? Is it linear interpolation between monthly-mean values centered at the 15th of the month, for instance?

Tables and Figures:

10. Table 2: Mention in the table caption the difference between stars (data available, but no SSW detected), and gray shading (no data available).

11. Figure 3: Mention contour levels for temperature anomaly in (a), and I'm guessing bold is 0 K.

---

## Referee Comment (RC2) · Anonymous Referee #2 · 14 Nov 2016

The authors provide a freely accessible and expendable compendium of sudden stratospheric warming events based on six meteorological reanalyses. Such a compendium will be of great value for the science community. The authors can be thanked for the effort to perform the work and to make it easily accessible.

The manuscript is well written. The data is well accessible and good documented. I have only one minor comment. The observational data base in the stratosphere before 1964 is rather sparse. In particular radiosondes in Russia seldom if any reached the stratosphere. I missed a related critical discussion. I recommend adding such a discussion. I expect that afterwards the early SSWs will be discarded or at least marked as problematic cases. I don't doubt that around the listed dates SSWs happened. However, I doubt, that the evolution is documented well enough in the reanalyses.

After the observational issue has been addressed properly I recommend publishing the manuscript.

---

## Author Comment (AC1) · 20 Dec 2016

**Final Author Comments**

We sincerely thank William Seviour and one anonymous reviewer for their constructive comments. Our response to the comments are indicated in Bold.

Response to RC1
The drop-down menu for downloading data at http://esrl.noaa.gov/csd/groups/csd8/sswcompendium/ appears not to be working, and only lists JRA-55 on the menu for reanalysis products. While data can be downloaded from the NCEI at https://www.ncei.noaa.gov/data/ssw/, the file names are quite cryptic. I would encourage the authors to ensure that the drop-down menu download is working, and include all of the fields in the SSWC. This would improve the accessibility of SSWC data, particularly for new users.

Data accessibility: **After careful consideration, we have decided to remove the option to download select data from the ESRL website. The website now directs the user to the NCEI archive site. The local ESRL website was never meant to host the full compendium, and we think it will be better if the NCEI is the sole source of the data files. We do apologize for the cryptic file names, but they were largely dictated to us by NCEI policies. We have provided an explanation for the file names in the User's Guide document. As far as the plotting functionality on the ESRL webpage, we have plans to add plots for the other reanalysis products in the near future. We have also added a link to a table of major SSWs, with plans to include links to animations of potential vorticity during these events. Hopefully these options will be useful for those users who just want a quick look at SSW events; and those who want the full data will have a consistent and well-maintained source of the files at NCEI.**

Comments:
1) Line 4/Page 2: Make clear the sign of the impact of SSWs on cold air outbreaks (i.e. "..., such as increasing the likelihood of cold air outbreaks ...")

**Changed text to: "These extreme events can have substantial impacts on wintertime surface climate, including increased frequency of cold air outbreaks over North America and Eurasia and anomalous warming over Greenland and eastern Canada."**

2) L21/P2: Change "40 degrees K" to "40 K", since Kelvin is absolute, not a 'degree'.

**Changed.**

3) L4/P5: "... in final warmings, the vortex breaks down but never recovers back to its climatological westerly state until the following boreal autumn": The climatological state following a final warming may actually be easterly (if it is a late final warming). I think saying "in final warmings, the vortex breaks down and becomes easterly until the following boreal autumn" is more accurate.

**Changed text to "... in *final warmings*, the vortex breaks down and remains easterly until the following boreal autumn."**

4) L12/P5: Perhaps mention the motivation for choosing 20 days between events. Radiative time scales?

**To clarify, changed text to: "The winds must return to westerly for 20 consecutive days between events (to avoid counting the same event twice; roughly equivalent to the thermal damping timescale at 10 hPa, Newman and Rosenfield 1997)."**

5) L9/P6: Total column ozone is not on pressure-levels as this implies, but is vertically integrated.

**Changed to: "We extracted the following fields (when available): vertically-integrated total column ozone; zonal winds, meridional winds, temperatures, geopotential heights, Ertel's potential vorticity (PV), and ozone mixing ratio, on provided pressure-levels; and at the surface, mean daily temperature, minimum daily temperature, maximum daily temperature, mean sea-level pressure, surface pressure, total precipitation liquid water equivalent, and total snowfall liquid water equivalent."**

6) L19/P6: Mention the value of c_p used for calculating isentropic levels.

**The coefficient kappa for calculating theta is defined to be (R/c_p). We used the values of these terms (287/1004) rather than the approximation. These values have been added to the text.**

7) L20/P6: Could a bit more information on the interpolation be provided? Was it linear interpolation?

**Linear interpolation is used for the interpolation to isentropes; this has been clarified in the text. The theta values on each pressure level are first calculated (as a function of the pressure level itself and of temperature). As one goes vertically, these theta values are monotonically increasing. The fields we interpolate to isentropes are then linearly interpolated from the calculated theta values on pressure levels to the desired theta levels.**

8) L9/P7: For data sets that do not provide daily maximum/minimum temperature, how accurate is the calculation from 6-hourly or 3-hourly data? Are the minimum/maximum 6/3-hourly values used, or is there interpolation between them to calculate these values?

**We do not interpolate between the data to find the daily maximum/minimum temperatures (spline interpolation, or other higher order interpolation, could approximate them, but they would be entirely an artifact of interpolation, not of data-bounded physics). Since these reanalyses don't explicitly provide**

**maximum/minimum temperatures, these are the best solution for the given data. We have made a note of this in the text.**

9) L24/P8 and L34/P8: How are monthly-mean values interpolated to give daily values? Is it linear interpolation between monthly-mean values centered at the 15th of the month, for instance?

**The monthly-mean values of these indices (QBO and SOI) are considered to be measured on the 15th of each month. These values are linearly interpolated to daily data. This has been added to the text.**

10) Table 2: Mention in the table caption the difference between stars (data available, but no SSW detected), and gray shading (no data available).

**Done.**

11) Figure 3: Mention contour levels for temperature anomaly in (a), and I'm guessing bold is 0 K.

**In Fig. 3a, the temperature anomaly contour spacing is 2 K. We have added this to the caption.**

Response to RC2

I have only one minor comment. The observational database in the stratosphere before 1964 is rather sparse. In particular radiosondes in Russia seldom if any reached the stratosphere. I missed a related critical discussion. I recommend adding such a discussion. I expect that afterwards the early SSWs will be discarded or at least marked as problematic cases. I don't doubt that around the listed dates SSWs happened. However, I doubt, that the evolution is documented well enough in the reanalyses.

**We have added the following text on page 4, which we hope addresses the reviewer's concerns. We are in overall agreement with the reviewer, but also do not want to simply discard those events, some of which have been documented in other literature (see Teweles and Finger 1958, Hare 1960, Finger and Teweles 1964) and may have historical importance.**

**"In addition, the evolution of SSW events prior to 1964, when concentrated efforts to observe the upper atmosphere using radiosondes and rocketsondes were begun in association with the International Years of the Quiet Sun (IQSY), should be viewed with skepticism. Even radiosonde measurements of the stratosphere were very limited during that time period, and so reanalysis fields may be almost entirely model-driven."**